# Use of Transcutaneous Electrical Nerve Stimulation (TENS) for the Recovery of Oral Function after Orthognathic Surgery

**DOI:** 10.3390/jcm11123268

**Published:** 2022-06-07

**Authors:** Alberto Cacho, Cristina Tordera, César Colmenero

**Affiliations:** 1Section of Orthodontics, Faculty of Odontology, Complutense University, 28040 Madrid, Spain; cristinatorderagon@hotmail.com; 2Maxillofacial Surgeon, Ruber Hospital, 28040 Madrid, Spain; colmeneroruiz@hotmail.com; 3BIOCRAN (Craniofacial Biology) Research Group, Complutense University, 28006 Madrid, Spain

**Keywords:** orthognathic surgery, oral function, TENS

## Abstract

The oral functions of patients are markedly diminished immediately after orthognathic surgery, and novel approaches are needed to accelerate their recovery. The aim of this study was to examine the usefulness of weekly applications of transcutaneous electrical nerve stimulation (TENS) for this purpose, based on the evidence of its effectiveness in other types of patients with muscle alterations. Maximum jaw opening, bite force, pain, and facial inflammation were compared between patients receiving TENS and those receiving sham-TENS for 30 min at baseline and weekly over a four-week period after orthognathic surgery and were also compared between the before and after of each procedure. TENS was applied at 220 Hz, applying the maximum intensity tolerated by each individual patient. The TENS procedure was identical for all patients, but the device was not turned on in the sham-TENS group. Patients were blinded to their group membership. Results were analyzed separately in skeletal class II and III patients. Improvements in jaw opening and inflammation were significantly greater in the TENS than in the sham-TENS group, attributable to the muscle relaxation achieved with the procedure. Research is warranted on the benefits of a more frequent application of TENS.

## 1. Introduction

One of the main challenges in orthodontics is the correction of malocclusion due to skeletal disorders. Various procedures are available to address these problems in younger patients who are still growing, but surgery remains the sole option to improve the skeletal position and size and achieve the desired facial aesthetics in adults [1]. The refinement of surgical techniques and the utilization of 3D imaging in surgical splint planning and design have allowed highly satisfactory and more predictable outcomes to be obtained [2]. Nevertheless, some surgical complications have yet to be resolved, including a 60–70% reduction in jaw opening during the immediate postoperative period, which hampers correct eating behavior and slows the recovery of patients. For instance, Teng [3] reported a reduction of 57.23 to 21.61 mm in maximum jaw opening during the first month post-surgery, besides postoperative limitations in lateral excursion, speed of movement, angle, and distance of condylar movements in protrusion. Ko [4] and Jung [5] also described a significant reduction in jaw opening during the first month post-surgery, followed by a progressive improvement until baseline values were recovered at 24 months.

After the initial studies by Bell [6] in the 1980s, the only rehabilitation measure was to give patients a list of opening and closing exercises to initiate after removal of the intermaxillary fixation at 6 weeks post-surgery. Twenty years later, Oh [7] added the 5 min application of ultrasound three times a week to a 20 min program of opening and laterality exercises to be performed three times a day. Novel techniques proposed for rehabilitation after orthognathic surgery include transcutaneous electrical nerve stimulation (TENS), using a small battery-operated device connected to electrodes placed on painful areas. This non-pharmacological and non-invasive technique is widely applied to relieve acute and chronic pain in multiple diseases by activating the inhibitory pathways of the midbrain and brainstem and thereby inhibiting the excitability of nociceptive neurons in the spinal cord [8].

Murine studies by Sjölund [9] demonstrated that low-frequency electro-acupuncture increased endorphin levels in cerebrospinal fluid and that endogenous opioids were released at both supraspinal and spinal level when certain frequencies were applied. Karina Sato [10] and other researchers later confirmed that electro-acupuncture at low (2 Hz) and high (100 Hz) frequencies promotes opioid formation in the cerebrospinal fluid of rats. Numerous authors [8,11,12,13,14,15,16,17,18,19], subsequently found TENS to be effective, with varying degrees of statistical significance, in patients with fibromyalgia, bruxism, postoperative processes, pregnancy, and neurological spasticity.

TENS has been used not only for pain relief but also for muscle relaxation, which is highly relevant to the recovery of jaw mobility. The electric stimulation triggers muscle contraction, promoting the entry of Na^+^ into nerve fiber membranes and the release of K^+^ and Ca^++^ ions. Electrolyte concentrations in muscle fiber membranes remain altered for some time after TENS therapy, reducing the conduction velocity and thereby promoting muscle relaxation. There is also an increase in blood flow and therefore in muscle temperature, activating the sodium/potassium pump to recover the ion concentrations lost during the electrical stimulus [16,20]. No data have yet been published on the usefulness of TENS in the setting of orthognathic surgery, prompting the present study. The main objective was to determine whether weekly TENS applications could increase the bite force and jaw opening in patients undergoing orthognathic surgery and decrease their pain and inflammation.

## 2. Materials and Methods

A single-blind randomized clinical trial was conducted in patients scheduled for orthognathic surgery, divided between experimental and control groups. Participants were selected by non-probabilistic sampling of consecutive patients until the estimated sample size was reached. All patients had a skeletal and facial deformity amenable to surgery after a pre-surgical orthodontic period, and all signed their informed consent after receiving an information sheet on the study, which was approved by the ethics committee of the hospital (San Carlos Hospital, Madrid, Spain). The inclusion criterion was the programing of bimaxillary or mandibular orthognathic surgery by the same surgeon in order to avoid interoperator variability. Exclusion criteria were surgery for temporomandibular disorders or more complex syndromes, the presence of muscle or nervous disorders or receipt of medication for such disorders, the impossibility of attending follow-up appointments, or the refusal of informed consent.

The sample size was estimated to obtain a statistical power of 80% with an alpha error of 0.05 and 95% confidence interval to detect a difference in jaw opening ≥2.5 mm, calculating a minimum sample size of 40 patients. Patients were randomly assigned to the experimental group (N = 23) for TENS application or the control group for sham-TENS (N = 24). Simple randomization was conducted by the investigator, who asked the patient to extract one ball from a vessel containing two different-colored balls that could not be seen; patients were blinded to their group assignation, as was the researcher responsible for data analyses. The flowchart of patient recruitment is depicted in Figure 1.

### 2.1. TENS Procedure

An Enraf Nonius S82 model TENS device (Enraf Nonius, Rotterdam, The Netherlands) was used, with a maximum frequency of 120 Hz and an intensity range of 0 to 99.5 mA. TENS electrodes (diameter 35–52 mm) were placed bilaterally on mandibular elevator muscles, on the superficial masseter muscle above the gonial angle, and bilaterally on the anterior temporal muscle, following the manufacturer’s instructions. The device was applied in an identical manner to all patients in both groups and kept in position for the same time period (30 min); however, the device was not switched on for the control group, and the stimulation intensity was adjusted for those in the experimental group to the maximum that did not cause discomfort or areas of contraction, maintaining this stimulation intensity and frequency throughout the 30 min session. Each participant underwent a weekly TENS or sham-TENS session on the same day of the week during a four-week period; appointments were scheduled so as to minimize any possible interaction among study participants.

### 2.2. Study Variables

Data were gathered from all patients on jaw opening, bite force, inflammation, and pain before surgery and at 7, 14, 21, and 28 days post-surgery, conducting measurements both before and after the TENS/sham-TENS session.

Maximum Jaw Opening was evaluated with patients seated vertically upright in the dental chair, using a digital dental caliber (Model R 100110, Mestra) to measure the maximum opening from the incisal margin of upper central incisors to the incisal margin of lower incisors, adding the amount (in mm) of overbite or subtracting the amount (in mm) of open bite in occlusion.

Bite force was measured using Dental Prescale Fuji film, which is formed by microcapsules that generate a chemical reaction under pressure, staining the contact area with a color density corresponding to the pressure applied [21,22,23,24,25]. A sheet of the film was placed between the occlusal surfaces of the two arches, and the patient was asked to bite as strongly as possible for 5 sec. This procedure was repeated three times, selecting the sheet with the best-defined tooth print to be photographed with a Canon EOS 500 camera (Canon, Tokio, Japan) (RAW format, F32, and annular Flash) at the minimum distance permitted by the 75-macro lens. The image was processed in a Mac computer (Apple, Cupertino, CA, USA) to obtain the color value according to the Cie L*a*b* (CIELAB) scale, which gives the color a numerical value based on the color-opponent theory that two colors cannot be red and green or yellow and blue at the same time, permitting the use of single values to describe red/green and yellow/blue attributes. L* indicates lightness and a* and b* are chromatic coordinates indicating positions between green and red and between yellow and blue, respectively. After observing a close correlation among the three variables (L*, a*, and b*), L* values were selected for analysis as having the greatest impact on color changes. The corresponding pressure units (Megapascals, MPa) were calculated according to Dental Prescale specifications, and the bite force (in Newtons [N]) was obtained by using the following formula:Bite Force = Bite Pressure (MPa) × mm^2^ print surface.

Pain perceived by patients while autonomously opening and closing their jaws was evaluated using a visual analog scale (VAS).

Facial inflammation was measured (in mm) with patients seated upright in the dental chair, using a soft ruler to obtain a horizontal measurement from the lower border of the earlobe to the corner of the mouth and a vertical measurement from the gonial angle to the outer canthus of the eye. The soft ruler was adapted to the contour of the patient’s face without exerting any pressure.

All of the above measurements provided numeric values for treatment as continuous quantitative variables.

### 2.3. Statistical Analysis

Descriptive statistics were conducted to calculate distributions, frequencies, means and standard deviations for sex, malocclusion, and the results for maximum opening, bite force, inflammation, and pain in each group (experimental and control). The Kolmogorov–Smirnov test was applied to check the normality of variable distributions and Levene’s test to check the equality of variances. The Student’s *t*-test was employed to evaluate the difference in each measurement between experimental and control groups. The paired Student’s *t*-test was used to compare measurements before and after TENS/sham-TENS in the same patient. Finally, the GLM repeated-measures procedure was applied to analyze the results for each group over time. SPSS 22.0 for Windows (International Business Machines Corporation (IBM), Armonk, NY, USA) was used for statistical analysis and an Excel spreadsheet served for the database. *p* ≤ 0.05 was considered significant in all tests.

## 3. Results

After application of the study eligibility criteria, the final study sample comprised 47 patients, 32 females (68.1%) and 15 males (31.9%), aged between 25 and 65 years. No patients were lost to the follow-up. The test group was formed by 23 patients who received TENS and the control group by 24 patients who underwent a sham-TENS procedure. Patients were skeletally classified as class II (23 patients aged 25–65 years, 19 females and 4 males; 11 in TENS group and 12 in sham-TENS group) or class III (24 patients aged 25–65 years, 13 females and 11 males; 12 in TENS group and 12 in sham-TENS group) because of the difference in surgical approach between these classes.

Results were obtained on the: (1) within-group changes in the time course of each variable (mean value) over the four-week study period; (2) changes in each variable between the before and after of TENS/sham-TENS; and (3) differences between TENS and sham-TENS groups in each variable.

Bite force (Figure 2).

In class II patients, the bite force decreased after surgery in the TENS and sham-TENS treated patients and slowly recovered until a bite force close to pre-surgery values was obtained at week 4, when there was no significant difference between TENS and sham-TENS groups. However, the bite force was greater in the control vs. experimental group at week 1 (*p* = 0.010) and week 2 (*p* = 0.035). A similar time course was observed in class III patients, with no significant difference between experimental and control patients at any time point post-surgery, although the TENS group did not fully recover bite force. In class II patients, superior bite force values were recorded after than before either TENS or sham-TENS from week 3 onwards.

Jaw opening (Figure 3).

In both class II and class III patients, a marked post-surgical reduction in jaw opening was gradually recovered by both groups, although the pre-surgery value was not achieved in either group. There was a significantly faster rate of recovery in the TENS vs. sham-TENS group. Among class II patients, the maximal jaw opening was 7 mm greater in the TENS group at 4 weeks, a significant difference (*p* = 0.012), although it was less than 50% of the original value in both groups at this time point. Among class III patients, a greater jaw opening was observed at 4 weeks in the TENS vs. sham-TENS group, but the difference did not reach statistical significance (*p* = 0.253). Although post-TENS/sham-TENS measurements were always higher than pre-TENS/sham-TENS measurements in all patients, the differences were not statistically significant.

Pain (Figure 4).

In both class II and III patients, pain perception increased during the first week post-surgery and then gradually decreased until virtually no pain was reported by either group at week 4. No statistically significant between-group difference was observed between TENS and non-TENS groups among class II (*p* = 0.065) or class III (*p* = 0.725) patients. In both class II and class III patients, pain perception was always lower after than before TENS or sham-TENS sessions except at week 2 in the class II TENS group, although the difference did not reach statistical significance.

Horizontal inflammation (Figure 5).

Among class II and III patients, both TENS and sham-TENS groups showed a significant post-surgical increase in horizontal inflammation that decreased over time. Among class II patients, inflammation was significantly (*p* < 0.001) lower in the TENS vs. sham-TENS group at week 2, but there was no significant between-group difference at week 4 (*p* = 0.386). Class II patients always showed a significant reduction in inflammation after the TENS session (especially at week 2), whereas there was no difference between the before and after of sham-TENS treatment at weeks 3 or 4. Among class III patients, horizontal inflammation was significantly lower at week 2 (*p* = 0.020) and week 4 (*p* = 0.030) in the TENS vs. sham-TENS group; however, there was no significant between-group difference (*p* = 0.137) in the reduction in inflammation over the study period.

Vertical inflammation (Figure 6).

In both class II and III patients, there was a post-surgical increase in vertical inflammation in both TENS and sham-TENS groups which was more marked at week 2 in class II patients. Vertical inflammation then gradually decreased in all study groups. Among class II patients, the reduction was significantly (*p* = 0.018) more rapid in the TENS group, and the vertical inflammation was 3.6 mm less at week 4 than before the surgery (baseline), whereas this value was the same as at baseline in the sham-TENS group. Among class III patients, vertical inflammation measured at week 4 was the same as at baseline, with no significant difference (*p* = 0.204) between TENS and sham-TENS groups. In the TENS group of class III patients, vertical inflammation was significantly less after than before the TENS application at weeks 1 (*p* = 0.021), 2 (*p* = 0.09), and 4 (*p* = 0.018).

## 4. Discussion

Orthognathic surgery has become increasingly predictable; however, concerns remain about the postsurgical rehabilitation of patients, with a need to accelerate the recovery of oral function. Various approaches are under investigation to meet this challenge, including ultrasound [5], manual lymphatic drainage [26], and the localized application of cold [27]. This study examined the effectiveness of TENS to improve jaw opening and reduce pain and inflammation in patients after orthognathic surgery, based on evidence of its effectiveness in other comparable situations [8,10,11,12,13,14,15,16,17,18,19,28].

Importantly, if the application of TENS proves to be useful, patients themselves will be able to use it for rehabilitation in their own homes. For this reason, conventional (not hospital) TENS was selected, with a frequency of 120 Hz, stimulus duration of 50 s, and a variable intensity as a function of the tolerance of patients, always within non-painful limits. The session duration was 30 min, following various authors [8,10,15,18,29]. Among all relevant studies consulted, the frequency has ranged from 100 to 120 Hz and the intensity has been 39.93 ± 13.79 mA. The TENS modality used combines conventional TENS, which stimulates A-beta fibers, with acupuncture-type TENS, which stimulates A-delta and C fibers.

The Dental Prescale Fuji system was selected as the most appropriate and precise approach to achieve bite force measurement. Electromyography [30], kinesiography [31], and gnathodynamometer [32] have been widely used for this purpose but are more complex and not suitable for weekly measurements or for patients with null mouth opening (immediately post-surgery). The T-Scan system is simple and appropriate but has shown poor reproducibility and quantification and is too rigid for patients in the immediate postoperative period [33]. Spectrophotometric methods with adenosine triphosphate or silicone adsorption are highly precise methods but difficult to reproduce, because patients must chew the material as if it were chewing gum; however, they are very reliable for healthy patients who have not undergone surgery that limits jaw movements [34,35]. A further advantage of the Dental Prescale system is that it yields simultaneous information on the force magnitude and the distribution of occlusal contacts, unlike kinesiographic and electromyographic methods. In comparison to T-Scan, it offers greater precision because it can measure force in teeth separated by <100 μm. Moreover, being a thin horseshoe-shaped film, it can be used easily in individuals with limited jaw opening, and its high degree of flexibility generates very precise dental prints [21,25,36,37,38]. Most studies using Dental Prescale Fuji for bite force measurement have used the Dental Prescale Occluzer system to evaluate prints. We followed authors such as Iwase [21] and measured the color intensity with the Cie L*a*b system, widely used in esthetic studies of tooth color [39,40]. It permits the specifying of color stimuli in a three-dimensional space.

The range of bite force values before surgery was 500–700 N, similar to findings by Iwase [21], Ohkura [25], and Nagai [36] who studied only class III patients after surgery, with no sham-surgery control group or rehabilitation measures.

Among class II patients, TENS application produced a significantly higher bite force post-TENS at weeks 1 and 2. This may be attributable to the well-documented effects of TENS application on muscle relaxation, which would reduce the perception of blockage by the patient and promote a more rapid recovery of bite force. In this way, the bite force at 4 weeks was greater in the TENS vs. sham-TENS group, similar to the results published before [21,25,36]. At this time point, pre-surgical bite force values were achieved by the TENS group but not by the sham-TENS group, although the between-group difference was not statistically significant. Among class III patients, there were no statistically significant differences in bite force between the TENS and sham-TENS groups at any time point.

Jaw opening was measured (in mm) with a digital caliper, a simple and rapid technique also used by various investigators [3,5,41,42]. In their studies of class III patients, Teng [3] and Ko [42] focused on jaw movements; therefore, our outcomes can only be compared with those of Jung [5] and Ueki [41], who reported similar jaw opening values of around 50 mm at baseline. After one month, the opening obtained by Ueki [41] was greater than in the present study (33 vs. 20.75 mm) whereas a similar jaw opening (22 mm) was observed by Jung [5]. Both class II and class III patients showed a gradual improvement after the major reduction that immediately followed the surgery, with a greater opening in the TENS vs. sham-TENS groups at week 4, although it remained only around half the jaw opening value recorded before the surgery. The opening was significantly greater (by 7 mm) in the TENS group among class II patients but not among class III patients.

The VAS used to record the pain experienced by patients is considered a solid, sensitive, and reproducible instrument and useful to re-evaluate pain in the same patient at different times [43]. Most of the class II and class III patients reported that the pain intensity was low throughout the process, decreasing after week 1 and returning to close to baseline values by week 4. Among class II patients, the pain was greater in the TENS vs. sham-TENS group throughout the study period, whereas the opposite was observed in class III patients until week 3. There is no evident explanation for these findings, although this subjective scale may be influenced by a desire to gratify the examiner, among other factors.

As expected, both horizontal and vertical inflammation values increased after surgery and then gradually decreased over time. Among class II patients, the TENS group had higher vertical and horizontal inflammation values in comparison to the sham-TENS group during the first two weeks and then showed a more marked improvement, obtaining a significant reduction in vertical inflammation by the end of the study period, when this reduction was 3.6 mm greater in the TENS vs. sham-TENS group. Among class III patients, the mean vertical inflammation was similar to baseline in both TENS and sham-TENS groups, with no statistically significant between-group differences. Among both class II and III patients, vertical and horizontal inflammation values were consistently lower after than before TENS but were also lower after than before the sham-TENS procedure in the first two weeks, which is difficult to explain.

Osunde [44], Oliveira Sierra [45], and Herrera Briones [46] measured inflammation after wisdom tooth extraction, using the distance between tragus-commissure and gonial angle-outer edge of the eye with a soft ruler in millimeters, as in our study. However, as is the case for pain, we have been unable to trace any study on facial inflammation after orthognathic surgery for comparison.

The results of this study cannot be extrapolated to other types of patients, and only one treatment regimen was employed, i.e., a single weekly session of 30 min. As a further limitation, the post-surgical elastic worn by the patients was replaced after each session, which is likely to have diminished the improvements in muscle relaxation, jaw opening, and inflammation achieved by TENS. Although patients in the control group would not have felt any stimulation (the device is turned off), they were always told by the operator that they would feel nothing during the procedure because the effect was produced by waves. For this reason, we do not consider that the experience would have revealed their group allocation. A strength of this study is that it offers the first report on the usefulness of TENS in patients undergoing orthognathic surgery, prompted by the lack of effective techniques to accelerate their post-surgical recovery. This simple device has been successfully used in other muscle diseases and can be applied by the patients themselves. This would even permit the daily application of this device, which could be expected to deliver greater benefits to patients. An additional study strength is the utilization of a single surgeon in all interventions, avoiding interoperator variability.

## 5. Conclusions

The weekly application of TENS for 30 min accelerates the recovery of oral function after orthognathic surgery. At four weeks, maximum jaw opening was significantly greater and vertical inflammation significantly less in skeletal class II patients receiving TENS than in those receiving sham-TENS treatment. Among class III patients, there appeared to be a greater improvement in jaw opening in those receiving TENS, but no statistically significant between-group differences were found. Further research is warranted on the effects of a more frequent application of TENS in these patients.

## Figures and Tables

**Figure 1 jcm-11-03268-f001:**
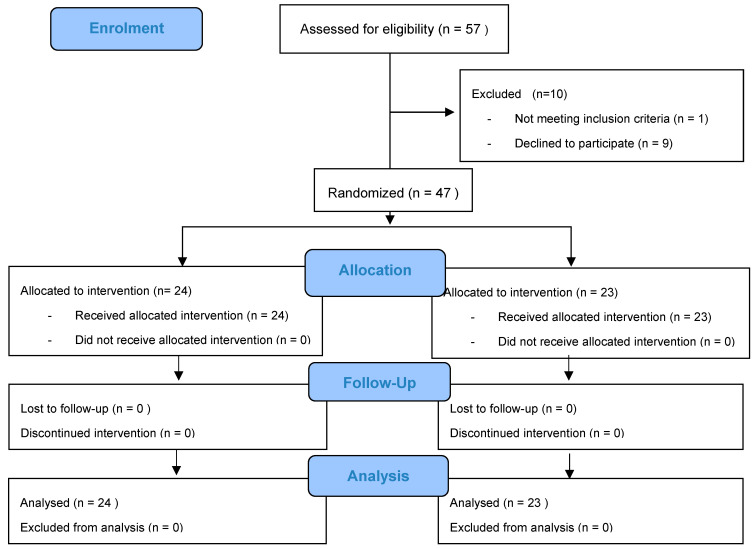
The flowchart of patient recruitment.

**Figure 2 jcm-11-03268-f002:**
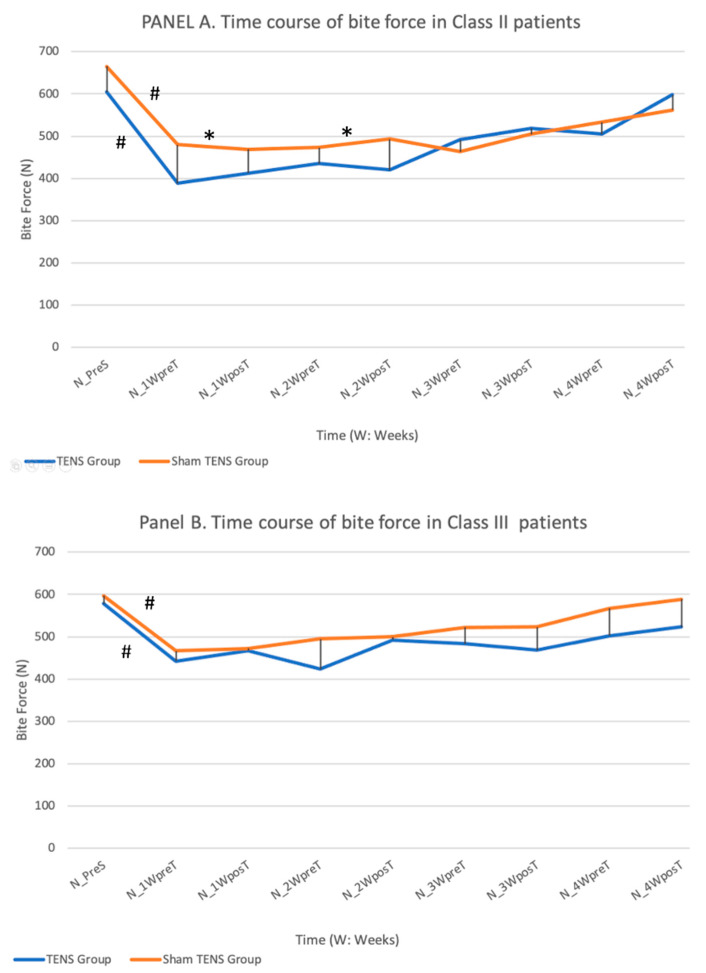
Depicts the changes in bite force after surgery in patients with skeletal class II (**A**) and III (**B**) receiving TENS and sham-TENS. (**#** significant differences (*p* < 0.05) between different time points in the same group, * significant differences (*p* < 0.05) between groups).

**Figure 3 jcm-11-03268-f003:**
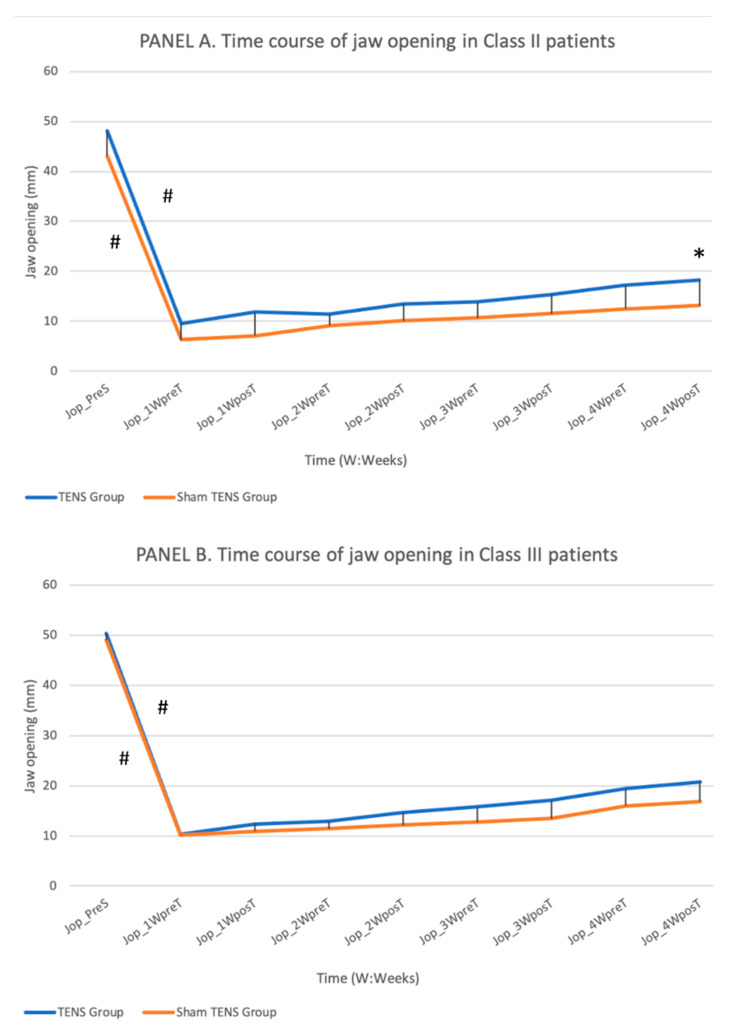
Depicts the time course of maximum jaw opening in the TENS and sham-TENS groups of class II (**A**) and class III (**B**) patients over the study period. (**#** significant differences (*p* < 0.05) between different time points in the same group, * significant differences (*p* < 0.05) between groups).

**Figure 4 jcm-11-03268-f004:**
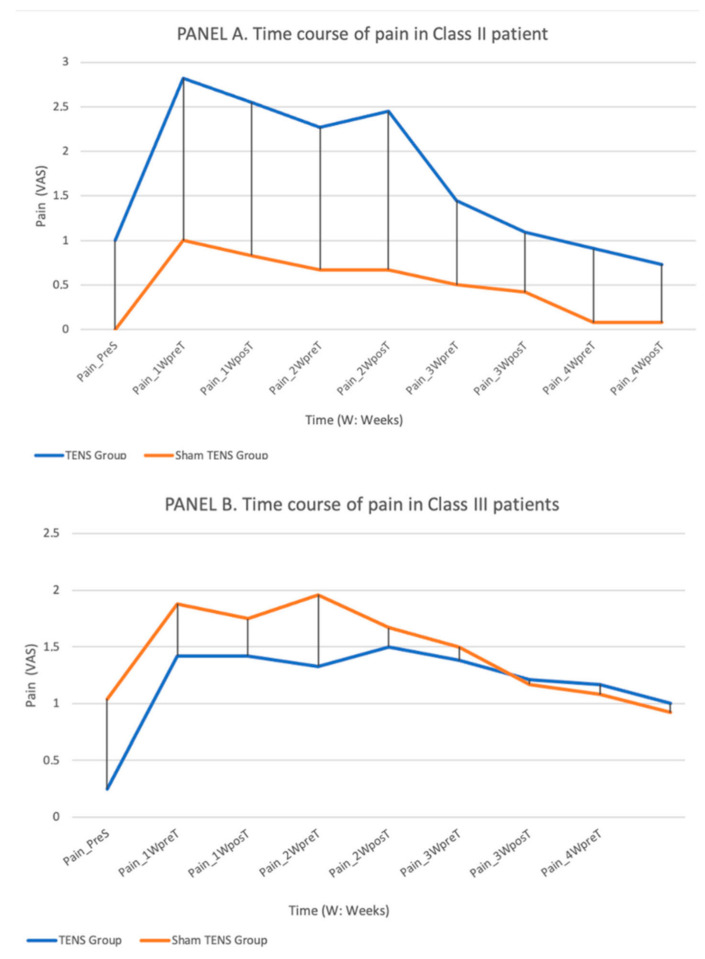
Depicts the time course of VAS pain values in the TENS and sham-TENS groups of class II patients (**A**) and class III patients (**B**) over the study period.

**Figure 5 jcm-11-03268-f005:**
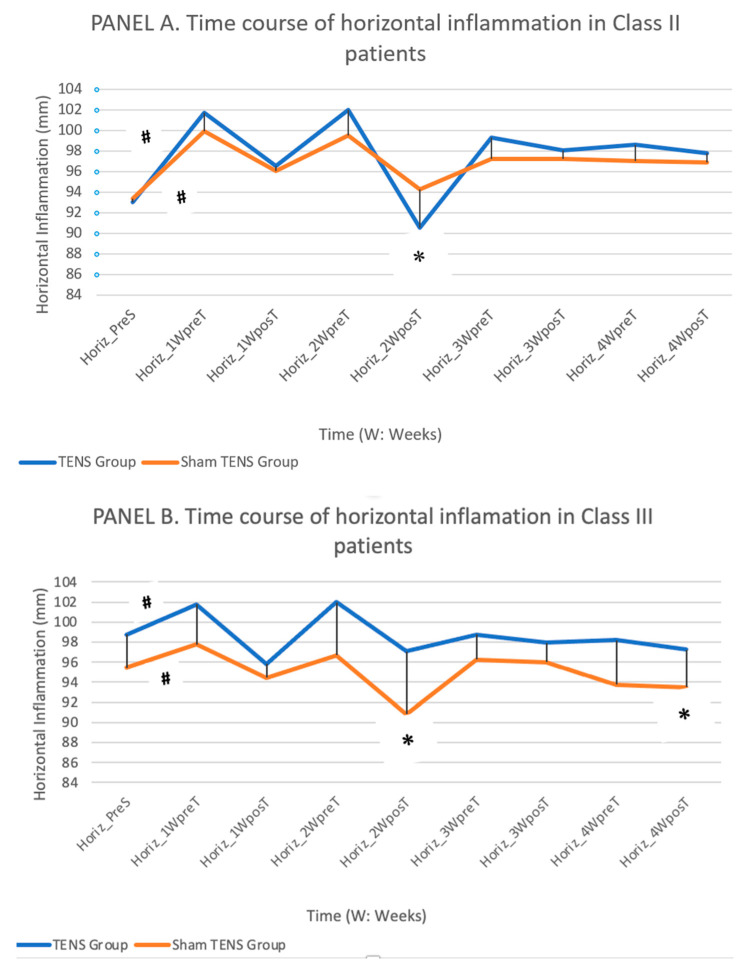
Depicts the time course of horizontal inflammation values in the TENS and sham-TENS groups of class II patients (**A**) and class III patients (**B**) over the study period. (**#** significant differences (*p* < 0.05) between different time points in the same group, * significant differences (*p* < 0.05) between groups).

**Figure 6 jcm-11-03268-f006:**
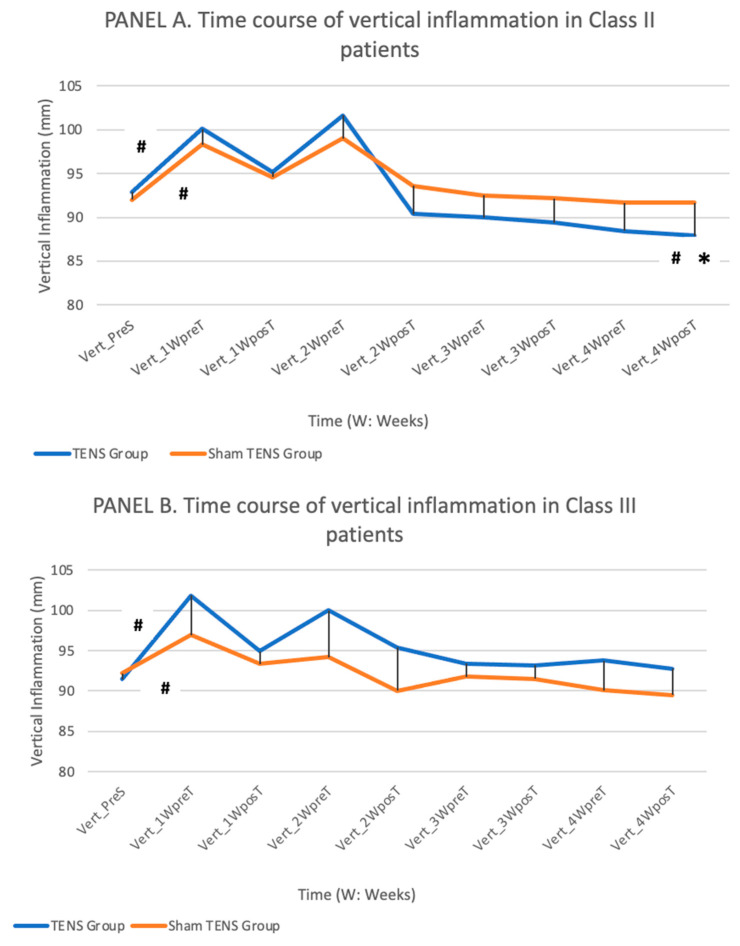
Depicts the time course of vertical inflammation values in the TENS and sham-TENS groups of class II patients (**A**) and class III patients (**B**) over the study period. (**#** significant differences (*p* < 0.05) between different time points in the same group, * significant differences (*p* < 0.05) between groups).

## Data Availability

We provide details regarding where data supporting reported results in the Appendix A.

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
