# Peer review of "Use of Transcutaneous Electrical Nerve Stimulation (TENS) for the Recovery of Oral Function after Orthognathic Surgery"

_jcm, 2022, doi:10.3390/jcm11123268_

Round 1

Reviewer 1 Report

Excellent work with a relevant topic. Authors should use the CONSORT guideline for writing therapy studies and send the checklist to authors to improve methods.
For example, add the flowchart, table 1 comparing both groups at baseline, say how the allocation concealment was performed, type of analysis, etc.
"Schulz K F, Altman DG, Moher D. CONSORT statement 2010: updated guidelines for reporting parallel-group randomized trials BMJ 2010; 340:c332 doi:10.1136/BMJ.c332

Author Response

RESPONSES TO REVIEWERS

Peer 1

Response: We are grateful for the observations of this reviewer, which have helped us to improve the quality of our paper.

We followed CONSORT guidelines and have provided a flowchart of patient recruitment (Table 1). We enclose the checklist for authors. The allocation concealment process is described in lines 89-90, and the type of analysis is reported in line 73.

We are very grateful to this reviewer for these valuable suggestions and insights, which have helped us to strengthen our paper.

Title and abstract

1a

Identification as a randomised trial in the title

NO

1b

Structured summary of trial design, methods, results, and conclusions (for specific guidance see CONSORT for abstracts)

1

Introduction

Background and objectives

2a

Scientific background and explanation of rationale

1-2

2b

Specific objectives or hypotheses

2

Methods

Trial design

3a

Description of trial design (such as parallel, factorial) including allocation ratio

2

3b

Important changes to methods after trial commencement (such as eligibility criteria), with reasons

NO

Participants

4a

Eligibility criteria for participants

2

4b

Settings and locations where the data were collected

2

Interventions

5

The interventions for each group with sufficient details to allow replication, including how and when they were actually administered

2-4

Outcomes

6a

Completely defined pre-specified primary and secondary outcome measures, including how and when they were assessed

3-4

6b

Any changes to trial outcomes after the trial commenced, with reasons

NO

Sample size

7a

How sample size was determined

2

7b

When applicable, explanation of any interim analyses and stopping guidelines

NO applicable

Randomisation:

 Sequence generation

8a

Method used to generate the random allocation sequence

2

8b

Type of randomisation; details of any restriction (such as blocking and block size)

2

 Allocation concealment mechanism

9

Mechanism used to implement the random allocation sequence (such as sequentially numbered containers), describing any steps taken to conceal the sequence until interventions were assigned

NO

 Implementation

10

Who generated the random allocation sequence, who enrolled participants, and who assigned participants to interventions

2

Blinding

11a

If done, who was blinded after assignment to interventions (for example, participants, care providers, those assessing outcomes) and how

2

11b

If relevant, description of the similarity of interventions

NO

Statistical methods

12a

Statistical methods used to compare groups for primary and secondary outcomes

4

12b

Methods for additional analyses, such as subgroup analyses and adjusted analyses

NO applicable

Results

Participant flow (a diagram is strongly recommended)

13a

For each group, the numbers of participants who were randomly assigned, received intended treatment, and were analysed for the primary outcome

4

13b

For each group, losses and exclusions after randomisation, together with reasons

3 (Table 1)

Recruitment

14a

Dates defining the periods of recruitment and follow-up

3 (Table 1)

14b

Why the trial ended or was stopped

the trial no ended or was stopped

Baseline data

15

A table showing baseline demographic and clinical characteristics for each group

NO

Numbers analysed

16

For each group, number of participants (denominator) included in each analysis and whether the analysis was by original assigned groups

4

Outcomes and estimation

17a

For each primary and secondary outcome, results for each group, and the estimated effect size and its precision (such as 95% confidence interval)

5-7

17b

For binary outcomes, presentation of both absolute and relative effect sizes is recommended

NO

Ancillary analyses

18

Results of any other analyses performed, including subgroup analyses and adjusted analyses, distinguishing pre-specified from exploratory

NO

Harms

19

All important harms or unintended effects in each group (for specific guidance see CONSORT for harms)

No harms or unintended effects

Discussion

Limitations

20

Trial limitations, addressing sources of potential bias, imprecision, and, if relevant, multiplicity of analyses

9

Generalisability

21

Generalisability (external validity, applicability) of the trial findings

9

Interpretation

22

Interpretation consistent with results, balancing benefits and harms, and considering other relevant evidence

7-9

Other information

Registration

23

Registration number and name of trial registry

NO

Protocol

24

Where the full trial protocol can be accessed, if available

San Carlos Hospital, Madrid, Spain

Funding

25

Sources of funding and other support (such as supply of drugs), role of funders

This research received no external funding

Reviewer 2 Report

Abstract

The first time that the authors write TENS, must be write after:  transcutaneous electrical nerve stimulation

Introduction

The introduction presents a good approach to the state of the art.

From lines 26 to 33 the references are missing.

The study aim should be more focused, referring to the primary outcome of the study.

Materials and Methods

Did they not record the study protocol? Why?

If you are reporting an RCT you should follow the CONSORT guidelines, using the flow chart and checklist of these guidelines.

Line 72: when they say it's double-blind who's blind besides the patients? Who did the TENS? Who evaluated the results?

Line 81: How was the presence of muscle or nerve disorders assessed?

Don't the authors think that the inclusion of patients who had only mandibular surgery and others who had bimaxillary surgery might influence the results?

Was there a control between how many patients of each type were included in each of the groups? Or when referring to classes II and III are they talking about this distinction? If that's the case, that's not clear.

Results

Figures 1,2,3,4 and 5: indicate in the figures the time-points where there are significant differences. For example with * to indicate between groups and # to indicate between different time-points in the same group.

Figure 4: I think edema would be a more appropriate term. Inflammation is a much more complex process and what was evaluated was edema.

Discussion

The authors should present the discussion without seccions

The application of TENS is perceived by patients who feel the stimulation. In this case, I do not think it is correct to say that the control group is blind. The patient feels whether he is being given TENS or not. Discuss this.

Lines 272-273: must modify the sentence. The authors cannot state that it was due to this, as these parameters were not directly evaluated in the study.

The discussion features a large repetition of the results which I suggest be removed. Authors should discuss potential explanations for the differences observed between the class II and class III groups.

Conclusions
The conclusions are misleading. The authors refer to differences in class II patients as significant when they are not. They must distinguish between statistically significant results and those that are not.

Author Response

RESPONSES TO REVIEWERS

Peer 2

Response: We are grateful for these comments on our preliminary attempt to improve the postoperative recovery of our patients. The reviewer will be interested to know that TENS is currently being applied daily by our patients at home, with highly encouraging results.

With regard to the English of the article, it has been thoroughly revised by Richard Davies MA, graduate of Cambridge University with more than 30 years’ experience preparing articles for publication in high-impact journals in the biohealth field. A certificate to this effect can be furnished if required.

ABSTRACT

  • The first time that the authors write TENS, must be write after:  transcutaneous electrical nerve stimulation

Response: This has been done

INTRODUCTION 

  • From lines 26 to 33 the references are missing.

Response: We now cite the following reference in support of these statements:

  1. Naran S; Steinbacher D; Taylor J. Current Concepts in Orthognatic Surgery. Plast Reconstr Surg 2018, 141(6), 925e-936e. doi 1097/PRS.0000000000004438
  2. Alkhayer A; Piffkó J; Lippold C; Segatto E. Accuracy of virtual planning in orthognatic surgery: a systematic review. Head Face Med 2020, 16(1), 34. Doi 1186/s13005-020-00250-2
  • The study aim should be more focused, referring to the primary outcome of the study.

Response: We have modified our description of the study objective accordingly:

 “The main objective was to determine whether weekly TENS applications can improve and accelerate the recovery of oral functionality in patients undergoing orthognathic surgery”

 MATERIALS AND METHODS

  • Did they not record the study protocol? Why?

Response: The clinical trial reported in this study was not registered. This is because the development of the trial preceded publication of the "Final Rule for Clinical Trials Registration and Results Information Submission" in 2017, since when trials at our School have been duly registered. See registration and results information submission requirements at  https://www.federalregister.gov/documents/2016/09/21/2016-22129/clinical-trials-registration-and-results-information-submission  in DATES section.

  • If you are reporting an RCT you should follow the CONSORT guidelines, using the flow chart and checklist of these guidelines.

Response: We have followed CONSORT guidelines. The corresponding flow chart has been introduced in the manuscript as table 1 and we have completed the checklist for authors.

  • Line 72: when they say it's double-blind who's blind besides the patients? Who did the TENS? Who evaluated the results?

Response: This was misleading and has been corrected to “single-blind”. We have added the following clarification in the revised text (lines 91-92):

“; patients were blinded to their group assignation, as was the researcher responsible for data analyses.“

  • Line 81: How was the presence of muscle or nerve disorders assessed?

Response: This was recorded when the medical records of the patient showed a history of muscle or nerve disorders.

  • Don't the authors think that the inclusion of patients who had only mandibular surgery and others who had bimaxillary surgery might influence the results?

Response: This an interesting question. In accordance with our study protocol, patients undergoing mandibular surgery were included, whether they also underwent maxillary surgery or not. We can offer no relevant data on this issue, which requires future investigation

  • Was there a control between how many patients of each type were included in each of the groups? Or when referring to classes II and III are they talking about this distinction? If that's the case, that's not clear.

Response: As stated in Results, there were 11 class II patients in the TENS group and 12 in sham-TENS group, and there were 12 class III patients in the TENS group and 12 in the sham-TENS group). We have checked that group and class are correctly specified in the Results and Discussion sections. It is now explained in the revised manuscript that patients were divided between these classes because of the difference in surgical approach (mandibular set back or BSSO) (lines 160-167).

RESULTS

  • Figures 1,2,3,4 and 5: indicate in the figures the time-points where there are significant differences. For example, with * to indicate between groups and # to indicate between different time-points in the same group.

Response: We are grateful for this suggestion, which has been followed.  

  • Figure 4: I think edema would be a more appropriate term. Inflammation is a much more complex process and what was evaluated was edema.

Response: Although there are components of edema and inflammation after the surgery, we agree that what remains after a few days and can last up to a month is edema. Nevertheless, we have used the same term employed by the only authors to have addressed this issue to date (according to our search of the literature): see references 44-46

DISCUSSION

  • The authors should present the discussion without sections

Response: This has been done

  • The application of TENS is perceived by patients who feel the stimulation. In this case, I do not think it is correct to say that the control group is blind. The patient feels whether he is being given TENS or not. Discuss this.

Response: We have now addressed this point in the following addition to the Discussion (342-346):

“Although patients in the control group would not have felt any stimulation (the device is turned off), they were always told by the operator that they would feel nothing during the procedure because the effect was produced by waves. For this reason, we do not consider that the experience would have revealed their group allocation.”

  • Lines 272-273: must modify the sentence. The authors cannot state that it was due to this, as these parameters were not directly evaluated in the study.

Response: We have modified this sentence accordingly, it now reads (292-294):

“This may be attributable to the well-documented effects of TENS application on muscle relaxation, which would reduce the perception of blockage by the patient and promote a more rapid recovery of bite force”.

  • The discussion features a large repetition of the results which I suggest be removed. Authors should discuss potential explanations for the differences observed between the class II and class III groups.

Response: We have made every effort to ensure that references to our own results in the Discussion are strictly necessary, minimizing repetition. It is possible that the distinct surgeries undergone by class II and III patients may explain the differences in outcomes observed. However, we do not have data to support any speculation on this issue, which we would prefer to avoid.  

CONCLUSIONS

  • The conclusions are misleading. The authors refer to differences in class II patients as significant when they are not. They must distinguish between statistically significant results and those that are not.

Response: These conclusions have been modified accordingly, specifying the differences that were statistically significant.

We are very grateful to this reviewer for these valuable suggestions and insights, which have helped us to strengthen our paper.

Round 2

Reviewer 2 Report

I still find the aim not very specific. When the authors talk about recovery of oral function, I consider it to be too vague.Authors should mention which functions exactly they want to assess (bite force, mouth opening amplitude, pain...). That is, the authors must say that they want to assess whether TENS increases the opening amplitude, or decreases pain, or increases bite force... Images are unreadable in this version. The authors pasted the statistical symbols on top instead of adding to the image itself which makes the images unreadable. Better quality images must be provided. On the legends of the images: they must indicate the value of p considered statistically significant.

Author Response

Dear Editor:

We are pleased to enclose our revised manuscript, which includes all improvements suggested by your reviewer. The objective of the study has been rewritten as recommended, and the quality of the figures and legends has been improved.

We are grateful to your reviewer for the valuable suggestions made, which have helped us strengthen our paper.

With thanks,

Sincerely yours,